# β-Cyclodextrin-Based Poly (Vinyl Alcohol) Fibers for Sustained Release of Fragrances

**DOI:** 10.3390/polym14102002

**Published:** 2022-05-13

**Authors:** Chengyuan Xing, Xia Xu, Lei Song, Xiaoling Wang, Bangjing Li, Kun Guo

**Affiliations:** 1College of Pharmacy, Southwest Minzu University, Chengdu 610041, China; xing919969255@163.com (C.X.); xxia1011@163.com (X.X.); scsonglei@163.com (L.S.); wxl3232@sina.com (X.W.); 2Chengdu Institute of Biology, Chinese Academy of Sciences, Chengdu 610041, China; libj@cib.ac.cn

**Keywords:** β-cyclodextrin, polyvinyl alcohol, wet spinning, functional fibers, fragrance retention

## Abstract

Poly (vinyl alcohol)/β-cyclodextrin (PVA/CD) composite fibers are prepared by wet spinning followed by hot stretching. XRD results show that β-CDs are in an amorphous state in fiber, and β-CD can help maintain the fibrous crystal that exists in the composite fiber. The DSC results show that the total crystalline ratio of the composite fibers decreased with the increase of β-CD. The as-prepared composite fibers were further crosslinked with glutaraldehyde (GA) to improve their usability. The crosslinked structure, together with amorphous β-CD, contributes to the loading and sustained release of fragrance molecules that were studied. The fragrance retention ratio of PVA/CD/GA is 55.63% and 48.25% for *cis*-jasmone and citronella, even after 25 days. The inclusion complexes of β-CD and fragrance molecules are confirmed by 2D-FTIR, which is responsible for the sustained release of fragrance. This study may contribute to the mass production of wearable long-term scented fabrics.

## 1. Introduction

Fragrance is always associated with relaxation and happiness. However, high concentrations of fragrance molecules may cause some side effects for surrounding people, such as allergic rhinitis, asthma, eczema, etc. [1]. Moreover, a fast loss of smell always drives customers crazy: the nature of fragrance molecules makes it hard to keep the molecules stable for a long time without further treatment [2,3]. Thus, fragrance release systems that can deliver fragrance molecules consistently are highly desired. To achieve high storage stability and the slow release of fragrance molecules, several fragrance delivery systems have been developed. Microcapsules and microspheres are two options for the controlled release of fragrance that have been widely studied in recent years [4,5,6]. Although they can bind aromatic molecules well, specific stimulating switches are required to release the bounding fragrance molecules. What is more, by the time the switch turns on, the release becomes uncontrollable and more prone to violent release [7,8].

β-Cyclodextrin (β-CD) and its derivatives are torus-shaped molecules with a large number of exterior hydroxyl groups, which are characterized by a hydrophobic inner cavity and hydrophilic outer wall [9]. Many pieces of literature have reported that β-CDs are able to form host-guest inclusion complexes with fragrance molecules, which makes them an ideal controlled release candidate for fragrance. Wu et al. [10] constructed self-dispersed β-cyclodextrin-coupled cellulose via chemically coupling cellulose and β-CD in NaOH/urea aqueous solution for sustained release of vanillin.

Among the various fragrance carriers, fabric is an ideal candidate. The sustained release of fragrance molecules through fabric has aroused the great interest of fabric engineers [11,12]. The incorporation of β-CD onto fibers/fabrics can be conducted by dyeing, printing, padding, spraying, coating, inkjet printing, impregnation, and grafting [13,14]. However, simple ways such as blending are more promising in mass production for the preparation of functional fibers. Although there are many meaningful works on preparing electrospinning functional fibrous membrane [15,16], it is hard to turn the fibrous membrane into fabric, and it is difficult to realize the large-scale production for their practical applications. Thus, the cyclodextrin-containing fibers with sustained-release properties prepared by simple but efficient methods are still being explored.

In this work, we constructed mass-produced PVA/CD composite fibers by wet spinning and used them for fragrance retention after post-treatment with glutaraldehyde (GA). PVA was chosen as the polymer matrix because of the excellent mechanical property of resultant fibers owing to the crystallization ability and its good compatibility. β-CDs were used as functional molecules, mixing them with PVA to encapsulate fragrance molecules. In order to verify the feasibility of its application in the field of fragrance retention, we studied the crystallization properties of these composite fibers and found that β-CD exists in an amorphous form in the fiber. GA was used to crosslink with β-CD and PVA to improve its useability. The final products were denoted as PVA/50CD, PVA/75CD and PVA/100CD, according to the mass ratios of PVA and CD 1.0:0.5, 1.0:0.75 and 1.0:1.0, respectively. Moreover, *cis*-jasmone and citronella were used as model molecules to study sustained-release properties. The results showed that compared with the control PVA/GA fiber without the addition of β-CD, PVA/CD/GA fibers have a better sustained-release property, and they can still maintain the “medium” of scent ingredients even after 25 days. This confirmed its potential for fragrance sustained-release applications in the textile field.

## 2. Materials and Methods

### 2.1. Materials

PVA1799 (average degree of polymerization, 1700), β-cyclodextrin, glutaraldehyde (25 wt% in water solution), ethanol, ammonium sulphate (NH_4_SO_4_, 95.0%), sodium sulphate decahydrate (Na_2_SO_4_•10H_2_O, ≥99.0%) and sulphuric acid (H_2_SO_4_, 98.0%) were purchased from Kelong Chemical Reagent Company of Chengdu. Citronella (95%) and *cis*-jasmone (94%) were purchased from Macklin. All reagents and solvents were purchased from Aladdin, Shanghai and used as received. Ultrapure water (UP water, resistivity > 18.0 MΩ·cm, 25 °C) was also used in this work.

### 2.2. Preparation of the PVA and PVA/CD Fibers

The functional fibers were prepared by wet spinning. Neat PVA spinning dope was prepared by dissolving PVA in water at 95 °C by mechanical stirring with a mass percentage of 19.0% (*w*/*w*). The PVA/CD dopes were prepared by dissolving PVA and CD in water at 95 °C with mechanical stirring. The mass percentage of PVA was 19% (*w*/*w*) the mass ratios of PVA and β-CD were 1.0:0.5, 1.0:0.75 and 1.0:1.0, named PVA/50CD, PVA/75CD and PVA/100CD, respectively. Wet spun fibers were conducted by a pilot-scale wet spinning machine provided by Sichuan University. The as-prepared initial fibers were hot-stretched at 1:2 and hot set at 180 °C to get the resultant PVA and PVA/CD fibers.

### 2.3. Preparation of the PVA/GA and PVA/CD/GA Fibers

The acetylated solution was prepared as follows: 200 mL of 10% (*w*/*v*) sodium sulphate solution and several drops of sulphuric acid were added to adjust pH to 6.0, and GA at a final concentration of 10–25 g/L was added to the solution. Then, an amount of PVA or PVA/CD fibers in a stretching state were immersed in the solution and the temperature was raised from room temperature to 60 °C. Then, the solution was kept at 60 °C for 20 min to obtain acetylated fibers. The prepared acetylated fibers were washed with water to remove sodium sulphate and unreacted GA to get the final PVA/GA and PVA/CD/GA fibers.

### 2.4. Shrinkage of PVA/75CD after Crosslinked by Different Concentration of GA

Fibers with a 50 cm length of PVA/75CD were selected to crosslink with different concentrations of GA solutions (10, 15, 20 and 25 g/L) and kept at 60 °C for 20 min. Additionally, the fibers were kept in a stretched state. Then, the fibers were washed with water and the length after acetylation was measured. Three repeated samples were taken for each group. The rate of shrinkage was conducted by the equation: Shrinkage = (the length before acetylation-the length after acetylation)/(the length before acetylation) × 100%.

### 2.5. Preparation of Chloroacetyl Substituted β-CD

Detailed experimental procedures and analytical data are shown as follows: (5.0 g, 4.4 mmol) β-CD was dissolved in 30 mL of DMF under stirring, and later, (0.49 g, 44 mmol) chloracetyl chloride was added to the system slowly to ensure that monosubstitution occurred. Then, we raised the temperature to 80 °C and reacted for 12 h. We placed the product into acetone to ensure complete precipitation. At last, the product was dissolved with methanol and washed with acetone. The product was dried in a drying box and was ready for wet spinning (yield: 62.9%); ^1^H NMR (400 MHz, DMSO-d_6_): δ 2.52 (m, 2H, Cl-CH_2_CO), 3.32–3.65 (m, H-2, H-3, H-4, H-5, H-6 of β-CD), 4.47 (m, OH-6 of CD), 4.85 (m, 7H, H-1 of β-CD), 5.70–5.78 (m, 14H, OH-2 and OH-3 of CD), 7.43, 7.75 (dt, 4H of phenyl).

### 2.6. Two-Dimensional Fourier Transform Infrared Spectroscopy (2D-FTIR Spectrum)

Two-Dimensional Fourier Transform Infrared Spectroscopy was used to confirm the inclusion of β-CD with *cis*-jasmone and citronella. For 2D-FTIR, the fibers were cut into pieces and made into a tablet with KBr. Then, the tablet was put into the sample pool with a temperature controller. Dynamic spectra were acquired at different temperatures ranging from 22 °C to 77 °C at a heating rate of 5 °C/min and kept for 3 min. Before the 2D correlation analysis, the experimental spectra were corrected by a linear baseline. The 2D correlation analysis was operated by the 2D-FTIR software programmed by Dr. Tao Zhou (Sichuan University, China). The 3% correlation intensity of the Proj-MW2D and null-space Proj-MW2D correlation spectra were removed, which was regarded as noise. Moreover, 1D-FTIR spectra were collected in the range of 4000–400 cm^−1^ with a resolution of 4 cm^−1^ and a total of 32 scans by an FTIR spectrometer (GX, Perkin-Elmer, CA, USA). Further, 1D-FTIR was processed by OMNIC 8.2 software.

### 2.7. Fragrance Retention Performance

The acetylated fibers were woven into fabrics (5 cm × 5 cm) by the knitting method. A total of 1.0 g *cis*-jasmone/citronella were dissolved in a solution of 2 mL of ethanol and 8 mL of water to prepare the solutions. Then, they were sprayed onto the fabrics to imitate the daily use of perfumes. The fabrics were kept at room temperature for 24 h, respectively, to ensure adequate host-guest inclusion. Then, the fabric was dried in a drying box at room temperature for three hours. The fabrics were placed on a well-ventilated platform at room temperature. The remaining *cis*-jasmone/citronella was extracted by 5 mL of methanol. The remaining perfumes were detected by high-performance liquid chromatography (Agilent 1200 HPLC). In both cases, the chromatographic separation was carried out with a ZORBAX Eclipse XDB C_18_ analytical column (25 mm × 4.6 mm i.d, 5.0 μm particle size) from Agilent (Agilent Technologies, Foster City, CA, USA). Mobile phases were 10% ultrapure water (eluent A) and 90% methanol (eluent B). 

A single-blind test was performed to subjectively characterize the fragrance retention by human beings. Briefly, a pool of trained groups, aged 20~30 years old, was asked to evaluate the rank of the four fabrics. The total number of experiences was 10, and at least 6 candidates in the pool were ensured to participate in each test. The odor intensity was divided into six grades:

0 No odor

1 Very weak

2 Weak

3 common

4 Strong

5 Very strong 

### 2.8. Characterization

A scanning electron microscope (SEM, JSM-7500F, JEOL, Tokyo, Japan) was used to characterize the morphology of the fibers. The accelerated voltage was 15 kV. The atomic distribution investigation (EDX mapping) was also conducted by SEM. A mechanical test was performed by an electronic single fiber strength tester (YG001A, HDFY, Beijing, China) with a drawing speed of 20 mm/min. Fourier transform infrared spectrophotometer (FTIR) was recorded using a Nexus-560 (Nicolet, USA). The wavenumber resolution was 2 cm^−1^, and the scan region was from 4000 cm^−1^ to 400 cm^−1^. X-ray diffraction (XRD) was carried out on a Bruker D8 Advance X-ray diffractometer (Bruker, Germany) using Cu Kα radiation in the scattering angle range of 2θ = 5–30° at a scan speed of 4° min^−1^. The DSC test was performed using a Q20 calorimeter (TA Instruments, New Castle, DE, USA). The temperature was varied from room temperature to 250 °C at a heating rate of 20 °C/min under nitrogen atmosphere. The degree of crystallinity of the samples was calculated from the ratio between the ∆Hm of the sample and the ∆Hm of 100% crystalline PVA (∆Hm = 150.0 J g^−1^). Melting enthalpy was derived from raw data from the DSC. Before the FTIR, XRD and DSC tests, the fibers were cut into powder and further results were obtained.

## 3. Results and Discussion

### 3.1. Preparation of PVA and PVA/CD Fibers

The preparation process is described in Figure 1a. In short, the spinning dopes were extruded through a spinneret, and the outer surface was solidified and rapidly formed a shell in a coagulation bath. As the moisture in the fibrous gel decreased, the formed shell collapsed under gravity. As solidification progressed, primary fibers were obtained [17]. In order to produce the fibers on a large scale and meet the need for commercialization, hot stretching was conducted, and the orientation of the molecular chain and the crystallization of the fibers were improved by hot stretching and hot setting (Appendix A). Figure 1b shows the SEM images of PVA and PVA/75CD fibers after the hot process. The pea-shaped cross-section fibers obtained are typical fabric fibers due to fast solidification.

Based on the FTIR analyses (Figure 1f), the characteristic peaks of CD, located at 1026 cm^−1^, 760 cm^−1^, are the vibration peaks of C–O–C skeleton and C–H. In addition, overlapping peaks of PVA and β-CD are 1648, 2945, and 3177–3420 cm^−1^, respectively, which are the stretching vibrations peaks of -OH and -CH. This verified the existence of β-CD within the fiber. Moreover, the hypochromic shift of -OH from 3208 cm^−1^ to 3177 cm^−1^ indicates the increased hydrogen bonding. This can be explained by the formation of a hydrogen bond between PVA and β-CD [18,19]. 

Figure 2a shows the XRD results of all four fibers. It can be seen that 11.4°, 19.5°, and 22.5° belong to the crystalline peaks of PVA. No characteristic crystalline peak of β-CD was found in all four samples, which proved the amorphous state of β-CD in the composite fibers. The crystallization peak at 20.2° is more pronounced with the increase of β-CD. This crystalline peak was also found in other highly oriented fibers and related to fiber processing [20,21]. For example, polyethylene fibers show fibrous crystals after stretching compared with unstretched fibers [22]. Therefore, the crystalline peak at 20.2° can be attributed to fibrous crystals and its existence can be related to stretching and β-CD. Firstly, the formation of the composite fiber without hot stretching is accompanied by the crystallization of PVA chains. This was also supported by the XRD results of the composite fibers that have not been hot-stretched (Figure 2b and Appendix A). Fiber solidification is also accompanied by crystallization. As the fiber stretched, its molecular chains are aligned to form fibrous crystals (Figure 2c). It is worth noting, however, that the fluidity of the β-CD in the spinning dope makes it easier to occupy the space in the spinning solution, thereby preventing the PVA chain from recovering to coil structure and maintaining the stretched chain structure more easily (Figure 2d). Therefore, the existence of β-CD helped to regulate the crystallization behavior of the PVA fibers.

Meanwhile, the space and interface effects of β-CD on the composite fibers were also studied. The polydispersion of β-CD also impedes the interference effect of the crystallization behavior, leading to the change of the full-width half maxima (FWHM) and crystalline dimension (Appendix A). The average crystalline dimension shows first an increase, and then a decrease with the increase of β-CD contents. The reason for this phenomenon can be explained by the small amount of β-CD, which is beneficial for PVA to form large-size fibrous crystals. Considering that the FWHM at 19.5° decreases after further increasing β-CD (Appendix A), both the crystalline region and the crystalline dimension (Appendix A) decrease due to the space and interface effects of β-CD for lamellae (Appendix A). In order to make a further realization of the effect of β-CD, we studied the crystallinity of fibers derived from DSC (Figure 2e). The crystallinity is obtained by the ratio of the enthalpy of fusion ∆Hm to the enthalpy of fusion of 100% crystalline PVA (150.0 J/g) (Appendix A). Increasing β-CD contents can induce a steric hindrance on the PVA matrix, therefore decreasing the total crystallinity degree of the composite fiber. The presence of amorphous β-CD also squeezed the space of the PVA lamella crystallization region, and the lamella crystallization ability is also affected. Since the melting point of different crystals differs, the varied exothermic peaks at different temperatures also indicate different kinds of crystals. The melting points of the new crystal peaks are 186.6 °C and 190.3 °C for PVA/75CD and PVA/100CD, respectively; these results were supported by the crystallization peak at 20.2° in the XRD results. The existence of β-CD can alleviate the coil behavior of the PVA chain, thus helping to maintain the fibrous crystal formed by long-chain PVA [23]. However, lamella, composed of short and medium molecular chains, is relatively stable in forming lamella. Furthermore, β-CD which is composed of seven D-glucopyranosyl units is a “short molecular chain” compared to the PVA chain, thus, it is more likely to disturb the lamella rather than a fibrous crystal. The total crystalline ratio of the composite fibers decreased with the increase of β-CD (Figure 2f).

Mechanical properties (Figure 2g) were tested by a single-fiber level test for more than ten fibers by using an electronic single fiber strength tester. The breaking strength of the fibers is provided by the PVA matrix, which is partially affected by β-CD. With the addition of β-CD increased, the breaking strength of the fiber decreased from 2.44 to 1.78 cN/dTex. This can be explained by the decrease in total crystallinity. The presence of β-CD can affect the lamella crystallization, thus the composite fibers showed a decrease in breaking strength with the increase of the β-CD content. On the other hand, the addition of β-CD can weaken the interaction between the polymer chains and increases the mobility of the polymer chain, thus reducing the strength and increasing the elongation at break. 

### 3.2. Fabrication of PVA/GA and PVA/CD/GA Fibers

The prepared PVA and PVA/CD fibers need to be formalized for further application. In this work, glutaraldehyde (GA) was used to crosslink with PVA and β-CD for its low cytotoxicity and its crosslinking feature compared with formaldehyde [24,25]. Meanwhile, the study of the crosslinking condition was also performed to get a satisfying sample (Appendix A). It should be noted, however, that the rate of shrinkage has decreased after increasing the GA concentration before 20 g/L. However, the performance dramatically falls when the GA concentration reaches 25 g/L and was difficult to keep its fiber formation. Thus, the concentration of GA for crosslinking was selected as 20 g/L. The schematic illustration of the crosslinking behavior is demonstrated in Figure 3a. Generally, aldehyde groups from GA in the presence of acid can acetylate with the hydroxyl group on PVA chains and β-CD. The final products can be acetal and hemal acetal, depending on the position of the neighboring hydroxyl group.

SEM images of the formalized fibers are shown in Figure 3b–e. Different from electrospinning PVA/CD fibers, which can only be crosslinked by the vapor of GA for at least 48 h, wet spun PVA/CD fibers can be crosslinked in water solution for only 1 h and still keep their initial morphology. This is because the acetylated fibers are kept in a stretching state and hardly swell compared to unstretched fibers. FTIR spectra of acetylated fibers are depicted in Figure 3f. The intensity of the peaks at 3200~3400 cm^−1^ from the PVA/75CD/GA fibers (area I) is not strong compared with PVA/CD (Figure 1f), and the peaks shifted from 3193 cm^−1^ to 3223 cm^−1^, indicating that the bonded hydroxyl group decreased. This can be explained by the -OH groups in β-CD participating in the crosslinking reaction. From area II and area III, the presence of C–H at 2850 cm^−1^ and C=O at 1710 cm^−1^ also verified the success of crosslinking.

Mechanical tests of formalized PVA/CD/GA fibers were also conducted to evaluate the effect of the crosslinking effect (Figure 4a). The stress at break was enhanced compared with the PVA/CD fibers, and the elongation at break decreased after crosslinking. This can be explained by the crosslinking effect that can strengthen the PVA network, but the crosslinking points can cause stress concentration, thus decreasing the elongation at break. The mechanical strength of the PVA/CD/GA fibers is 1.34–2.11 cN/dTex, compared with the mechanical strength of regenerated bacterial cellulose fibers within the range of 0.5~1.5cN/dTex, indicating the good mechanical property of the PVA/CD/GA fibers, and the potential in the fabrication of fabrics.

We further tested the water resistance property of the PVA/CD/GA fibers. Chlorinated CD (Cl-CD) has been synthesized to simulate the wash durability of the as-prepared PVA/CD/GA fibers. Figure 4b,c are EDS mappings of the prepared PVA/75Cl-CD/GA fibers before and after 10 wash cycles. The red dot in the two figures represents element Cl representing the content of β-CD. We can see that the red dots show a little decrease after 10 wash cycles. The results show that the as-prepared formalized fibers are relatively stable to undergo at least 10 wash cycles.

### 3.3. Fragrance Sustained-Release Ability of PVA/CD/GA Fibers

The PVA/CD/GA fibers were further utilized for fragrance sustained-release application. In this work, *cis*-jasmone and citronella were used as model molecules to characterize their fragrance retention properties. The two fragrance molecules can be included in the cavity of β-CD. This host-guest inclusion behavior provided a possibility for the delayed fragrance release compared to the microcapsules, which should be triggered by a proper stimulus.

The robustness and general applicability of the formalized fibers were further demonstrated by monitoring the release rates (Figure 5a,b) of *cis*-jasmone and citronella. Further, *cis*-jasmone diminished in a controlled manner, reaching 42.44%, 52.68% and 55.63% after 25 days for PVA/50CD/GA, PVA/75CD/GA and PVA/100CD/GA, respectively. Whereas, *cis*-jasmone from the control fiber PVA/GA cannot be detected on the 10th day. The same trend can be found in citronella; PVA/CD/GA fibers keep a retention rate of 35.71%, 38.27% and 48.25%, respectively, with the increase of β-CD contents. The PVA/GA fibers cannot be detected on the 15th day. These results show the fragrance retention of PVA/100CD/GA kept above a moderate level (>40%) after 25 days. Moreover, customer judgements (single-blind tests) were also conducted to verify its fragrance retention property. The fragrance molecules loaded onto PVA/CD/GA fibers indicated a lasting aroma at least for 25 days (Figure 5c,d), and the density of the three fibers remains above “common” for 25 days. The aroma intensity of all the four fibers was “strong” from the beginning. After 25 days, the aroma intensity of PVA/GA fiber was “very weak”, and the others were able to keep a “modern” level. However, compared to the HPLC results, results from PVA/GA fibers revealed that errors exist for customers to distinguish the weak smell. The single-blind test showed good fragrance retention property toward PVA/CD/GA fibers. Overall, the single-blind test, together with HPLC results, confirmed that the PVA/CD/GA fibers showed good fragrance retention properties.

### 3.4. Proposed Mechanism for Fragrance Retention

From Figure 5a,b, the control fibers PVA/GA show a fast release of fragrance, which ended at 5 days and 10 days, respectively, indicating that the crosslinked PVA fiber shows little fragrance retention property. For PVA/CD/GA fibers, the crosslinked fibers include crosslinked PVA and crosslinked β-CD. Because the crosslinked PVA matrix shows little fragrance retention property, the sustained release property should be mainly attributed to crosslinked β-CD. With the additional amount of β-CD increased, the retention rate increased. This can be explained by the host-guest inclusion ability of β-CD to encapsulate and stabilize the fragrance molecules in its hydrophobic cavity.

The interactions of β-CD and fragrance molecules were confirmed by 2D FTIR, which has been proved a powerful tool for studying the host-guest inclusion in many systems. Figure 6 and Appendix A show the 2D FTIR results of the PVA/75CD/GA fibers with *cis*-jasmone/citronella host-guest inclusion complex and their simple blends. Appendix A shows 1D-ATR-FTIR spectra of the PVA/75CD/GA fibers with *cis*-jasmone/citronella host-guest inclusion complex and their simple blends. From Figure 6a, we can see the typical peaks of β-CD (755 and 705 cm^−1^) and *cis*-jasmone (1150 cm^−1^, interconversion of ketone and enol) showed a correlation peak, indicating that the five-membered ring was included in the cavity of β-CD (Figure 6c). Interestingly, there is no typical correlation peak available in the blend sample. From Figure 6d, the typical peak of β-CD (755 and 705 cm^−1^, -OH) and citronella (1190 cm^−1^, -CO) also showed a correlation peak (Figure 6f), however, a similar correlation peak also cannot be found in CD and the citronella blend (Figure 6e). According to the theory for 2D IR spectroscopy developed by Noda [26], a correlation peak is generated when two dipole transition moments associated with molecular vibrations of different functional groups are reorienting simultaneously. Such cooperative motion of the local structure is expected when strong interactions exist among different groups. These results indicated that the strong interaction between β-CD and citronella is because of the formation of host-guest inclusion complexes. Thus, we can conclude that the fragrance molecules in the fibers are divided into two parts: one in the cavity of β-CD and the other in the crosslinked structure. Fragrance molecules in the crosslinked structure release fast for its porous structure and the molecules trapped in β-CD release at a low rate.

## 4. Conclusions

A series of PVA/CD/GA fibers have been prepared by wet spinning; the crystallization phenomenon has been discussed and its possible mechanism has been proposed. Furthermore, the wet spun fibers were formalized with GA to crosslink β-CD with PVA. The β-CDs on the fibers were responsible for the long-term sustained release of fragrance molecules, because of the ability to form a host-guest inclusion complex with fragrance molecules. The fragrance retention rate of PVA/CD/GA was as high as 55.63% and 48.25% for *cis*-jasmone and citronella after 25 days. This work demonstrates a large-scale preparation of the long-term fragrance retention fibers and shows potential in the fragrance retention of wearable fabrics. 

## Figures and Tables

**Figure 1 polymers-14-02002-f001:**
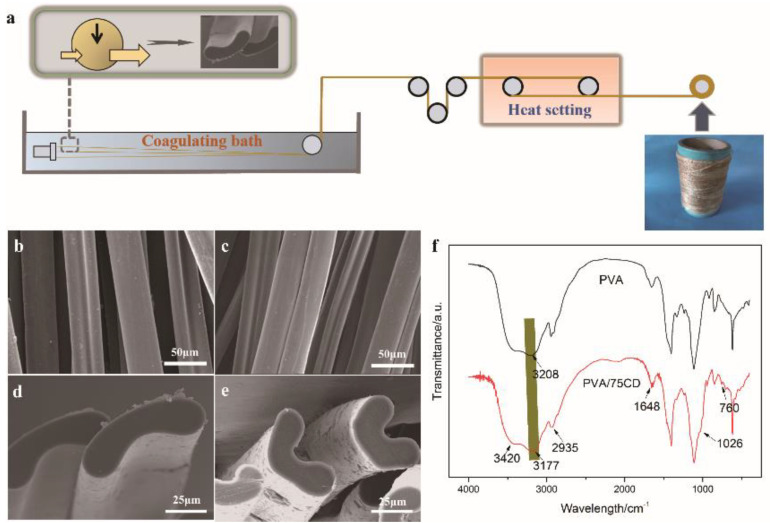
(**a**) Schematic illustration of the preparation of the composite fibers; (**b**). SEM image of the surface of PVA fibers; (**c**) SEM image of the surface of PVA/75CD fibers; (**d**) SEM image of the cross-section of PVA fibers; (**e**) SEM image of the cross-section of PVA/75CD fibers; (**f**) FTIR spectra of PVA and PVA/75CD fibers.

**Figure 2 polymers-14-02002-f002:**
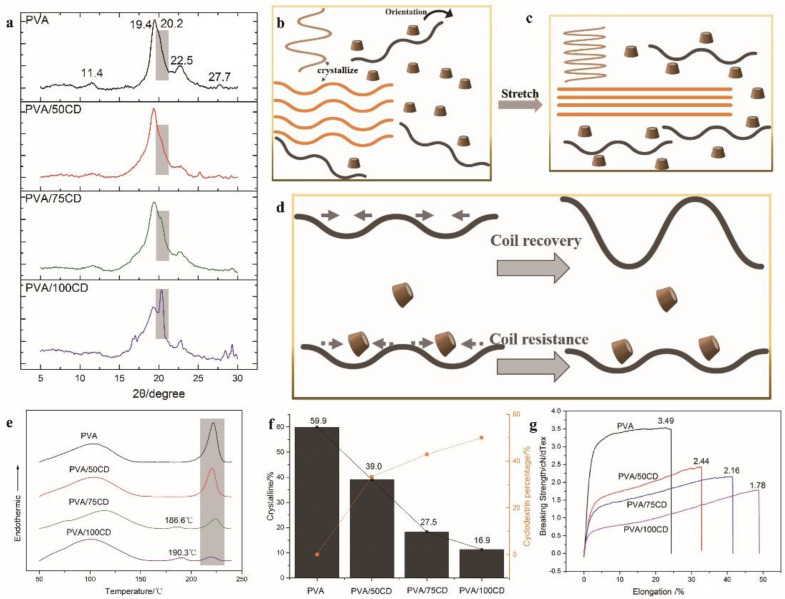
(**a**) XRD results of PVA, PVA/50CD, PVA/75CD and PVA/100CD fibers; illustration of the structure of (**b**) PVA/CD fibers before hot stretching, (**c**) after hot stretching and (**d**) coil resistance behavior; (**e**) DSC results of PVA, PVA/50CD, PVA/75CD and PVA/100CD fibers; (**f**) crystallinity ratio of PVA, PVA/50CD, PVA/75CD and PVA/100CD fibers; (**g**) mechanical properties of PVA, PVA/50CD, PVA/75CD and PVA/100CD fibers.

**Figure 3 polymers-14-02002-f003:**
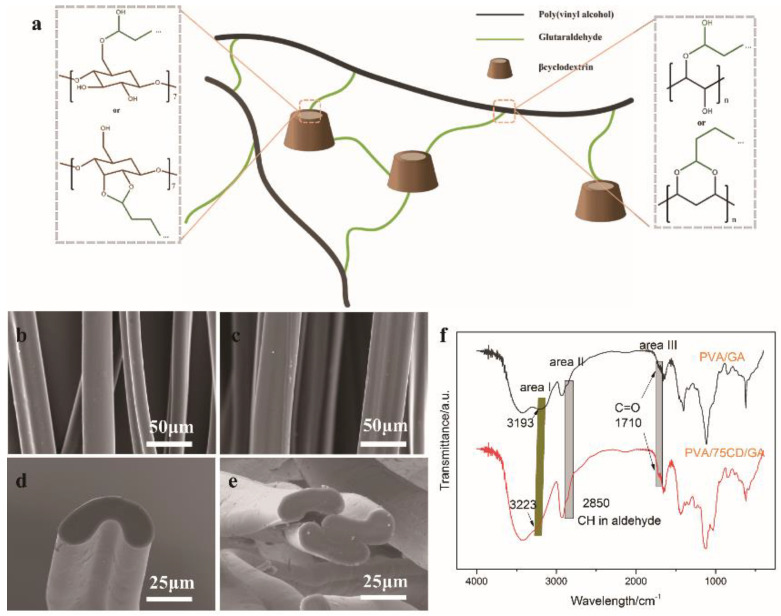
(**a**) Schematic illustration of PVA/CD fibers crosslinked with GA; (**b**) SEM images of PVA/GA fibers; (**c**) SEM images of PVA/75CD fibers; (**d**) cross-section SEM images of PVA/GA fibers; (**e**) cross-section SEM images of PVA/75CD/GA fibers; FTIR of PVA/GA and PVA/75CD/GA fibers; (**f**) FTIR spectra of acetylated fibers.

**Figure 4 polymers-14-02002-f004:**
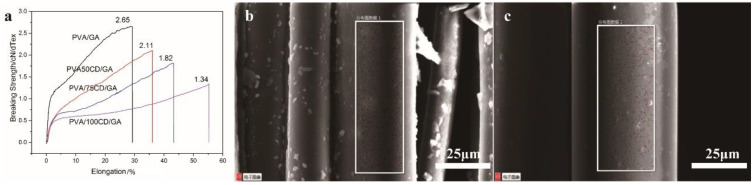
(**a**) Mechanical properties of PVA/GA, PVA/50CD/GA, PVA/75CD/GA and PVA/100CD/GA fibers; EDS mapping of Cl substituted PVA/75CD fiber before (**b**) and after (**c**) 10 wash cycles.

**Figure 5 polymers-14-02002-f005:**
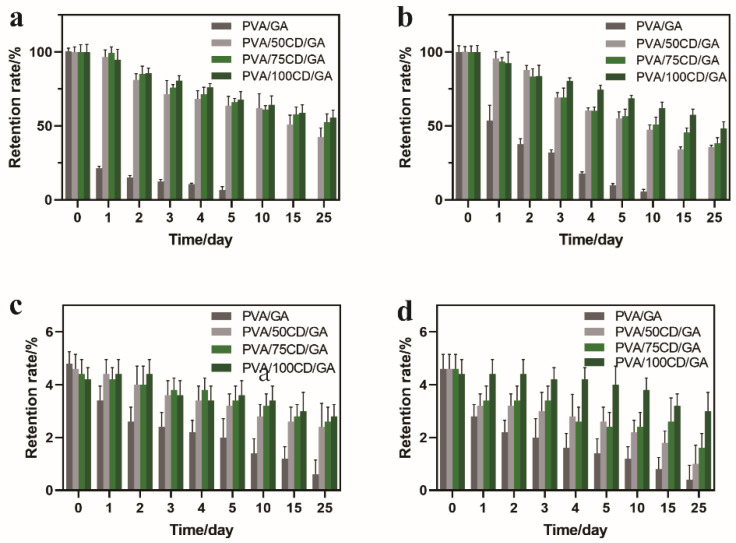
Fragrance retention HPLC results of *cis*-jasmone (**a**) and citronella (**b**) for PVA/GA, PVA/50CD/GA, PVA/75CD/GA and PVA/100CD/GA fibers; single-blind result of *cis*-jasmone (**c**) and citronella (**d**) for PVA/GA, PVA/50CD/GA, PVA/75CD/GA and PVA/100CD/GA fibers.

**Figure 6 polymers-14-02002-f006:**
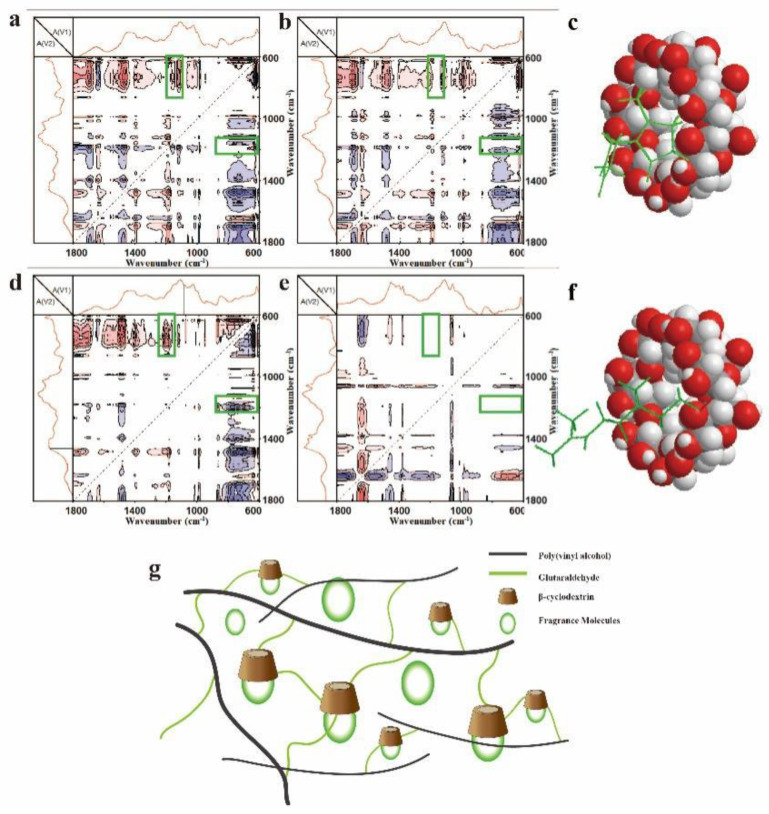
2D Asynchronous FTIR spectra calculated from the temperature-dependent spectra of 25–77 °C in the range of 1800–600 cm^−1^. (**a**) β-CD-*cis*-jasmone host-guest inclusion complex; (**b**) β-CD-*cis*-jasmone blend; (**c**) Interaction of β-CD with *cis*-jasmone; (**d**) β-CD-citronella host-guest inclusion complex; (**e**) β-CD-citronella blend. (**f**) Interaction of β-CD with citronella. Pink and blue areas represent the positive and negative correlation intensity, respectively. (**g**) Illustration of the distribution of fragrance molecules in PVA/CD/GA fibers.

## Data Availability

The data that supports the findings of this study are available from the corresponding author upon reasonable request.

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
