# Peer review of "β-Cyclodextrin-Based Poly (Vinyl Alcohol) Fibers for Sustained Release of Fragrances"

_polymers, 2022, doi:10.3390/polym14102002_

Round 1

Reviewer 1 Report

The title ‘β-Cyclodextrin based Poly (vinyl Alcohol) Fibers for Sustained Release of Fragrances’ can be rewritten as ‘β-Cyclodextrin-based Poly (vinyl Alcohol) Fibers for the Sustained Release of Fragrances’

Starting from the beginning, the English seems to be very weak and the whole manuscript needs major English editing.

Figure 5 is without statistical analysis.

Figure 6 is not clear. It should be replaced with a better quality one.

Information is missing regarding the details of XRD and DSC studies. The instrument details, methods etc.

Out of 39 references, 29 are in the introduction section. Unnecessary citation should be avoided. Some extra references should be deleted from the introduction.

What was the control used in Figure 5?

Overall the scope and idea of the study is nice, but the manuscript needs major improvements in writing, English and presentation of results. The figure quality should be improved throughout the manuscript.

Reviewer 2 Report

Dear Authors

The authors successfuly developed β-Cyclodextrin based Poly (vinyl Alcohol) Fibers for Sustained 2 Release of Fragrances. The research desgin is appropiate and the characterization used tools are adueqate. The presented results are interesting for the readers and sound.

However, some characterization measurements are still missing and further optimization work is needed.

The developed polymer matrices controlled release the Fragrances, so the porosity characters including pores size, pores size distribution, and finally pores volume. Moreover, the surface area needs to be measured.

All the abovementioned tests need to be performed and the obtained results need to be corellated to the polymer matrix composition and accordingly to the release behaviour. 

The authors proved the role of GA crosslinked the PVA and Beta-cyclodextrine, so the study of the GA concentration and the crosslinking process parameters is essential.

In conclusion, I can recommend your manuscript after performing the required revision. 

Round 2

Reviewer 1 Report

The authors have made some changes in the manuscript. Though, they are not major, still the manuscript is in better shape now.

Reviewer 2 Report

Dear Authors

Thanks for your consideration the comments during your revision.

I can recommend the revised manuscript for publication.